# Nutritional sex-specificity on bacterial metabolites during mosquito (*Aedes aegypti*) development leads to adult sex-ratio distortion
Ottavia Romoli[1,2,4], Javier Serrato-Salas[1,4], Chloé Gapp [1], Yanouk Epelboin[1], Pol Figueras Ivern[1], Frédéric Barras[3] & Mathilde Gendrin [1] ✉

Mosquitoes rely on their microbiota for B vitamin synthesis. We previously found that *Aedes aegypti* third-instar larvae cleared of their microbiota were impaired in their development, notably due to a lack of folic acid (vitamin B9). In this study, we found that diet supplementation using a cocktail of seven B vitamins did not improve mosquito developmental success, but rather had a significant impact on the sex-ratio of the resulting adults, with an enrichment of female mosquitoes emerging from B vitamin-treated larvae. A transcriptomic analysis of male and female larvae identified some sex-specific regulated genes upon vitamin treatment. When treating germ-free larvae with individual B vitamins, we detected a specific toxic effect related to biotin (vitamin B7) exposure at high concentrations. We then provided germ-free larvae with varying biotin doses and showed that males are sensitive to biotin toxicity at a lower concentration than females. Gnotobiotic larvae exposed to controlled low bacterial counts or with bacteria characterised by slower growth, show a male-enriched adult population, suggesting that males require less bacteria-derived nutrients than females. These findings indicate that during larval development, mosquitoes have sex-specific nutritional requirements and toxicity thresholds, which impact the sex ratio of adults.

*Aedes aegypti* mosquitoes are vectors of pathogens responsible for various human diseases, notably including yellow fever or dengue fever. They are predominantly concentrated in tropical and neotropical regions, posing a significant threat to approximately 3.9 billion people who could potentially contract dengue[1]. Mosquitoes are holometabolous insects, undergoing a complete metamorphic transformation from aquatic larvae to terrestrial adults. Consequently, several adult physiological traits, such as size and lifespan, heavily hinge on the quality of the insect larval development, specifically influenced by the nutritional status during this developmental stage.

Mosquitoes depend heavily on their microbiota for vital nutrients crucial for their larval development. In their natural habitat, mosquito larvae primarily feed on microorganisms, particulate organic matter, and detritus present in their breeding sites[2]. In controlled laboratory rearing environments, the main food sources for larvae typically comprise fish food or commercial pet food designed for rodents, cats, or dogs. Mosquito larvae hatched from microbe-free eggs and maintained in sterile conditions are halted in their development when provided with a sterile conventional diet[3]. However, their development can be rescued when a live microbiota is introduced or when provided nutrient-rich diet and kept in the dark, strongly implying that the microbiota plays a fundamental role in furnishing to its mosquito host essential nutrients that are not present in the diet[3–6]. These essential nutrients encompass critical elements such as essential amino acids, nucleosides, and B vitamins, which are beyond the synthetic capabilities of most insects[4,6]. The investigation of mosquito nutritional requirements and how the microbiota affects larval development could unveil crucial metabolic insights that might serve as the foundation for innovative vector control strategies.

To unravel the mechanisms behind the intricate interactions between mosquitoes and their microbiota, we previously developed a method that

[1]Microbiota of Insect Vectors Group, Institut Pasteur de la Guyane, Cayenne, French Guiana. [2]Viruses and RNA interference Unit, Institut Pasteur, Université Paris Cité, CNRS UMR3569, Paris, France. [3]SAMe Unit, Department of Microbiology, Institut Pasteur, Université Paris Cité, CNRS UMR6047, Paris, France. [4]These authors contributed equally: Ottavia Romoli, Javier Serrato-Salas. ✉e-mail: mathilde.gendrin@pasteur.fr

enables the generation of germ-free mosquitoes at virtually any stage of their development. Our approach consists in the colonisation of *Ae. aegypti* mosquitoes with an *Escherichia coli* strain auxotrophic for two amino acids, d-Alanine (D-Ala) and meso-diaminopimelic acid (*m*DAP), that are bacteria-specific and crucial for bacterial cell wall synthesis. When bacteria are supplied alongside D-Ala and *m*DAP to germ-free larvae, mosquito development is rescued. As soon as the larval rearing medium is changed to sterile water deprived of bacteria and of D-Ala and *m*DAP, bacterial growth is arrested and germ-free larvae are obtained. Using this method, we have been able to pinpoint folic acid as one of the critical metabolites furnished by the microbiota and required during mosquito larval development[4].

Vitamins of the B group are cofactors carrying on numerous cellular processes, encompassing essential functions in the electron transport chain (riboflavin, nicotinic acid), amino acid metabolism (pyridoxine, folic acid), lipid metabolism (biotin, riboflavin, nicotinic acid), and nucleic acid metabolism (folic acid, nicotinic acid)[7]. They are required in very low amounts, as they serve as coenzymes and are not consumed by the enzymes for which they act as cofactors[8]. Insects depend on their food and microbiota to supply B vitamins as they lack the complete metabolic pathways to synthesise them[9]. The insect model *Drosophila melanogaster* requires all B vitamins for optimal larval development[10] and similar necessities have been hypothesised for mosquitoes. As a matter of fact, B vitamins have been identified, along with sterols, amino acids and nucleosides, as compounds in synthetic diets designed for rearing mosquito larvae in the absence of a microbiota[11–13]. A more recent study has highlighted the critical roles played by amino acids and B vitamins in mosquito larval growth. Notably, riboflavin has been shown to be required throughout the entire larval development as it is light sensitive, and thiamine, pyridoxine, and folic acid were found to be specifically required by *Ae. aegypti* larvae to initiate pupation[6].

In our previous study, we found that folic acid provision partially rescued the development success of mosquito larvae deprived of their microbiota during late instars[4]. Here, we tested the effect of supplying different B vitamin doses on *Ae. aegypti* larval development. Larvae cleared of their microbiota during their third instar and supplemented with increasing amounts of B vitamins did not show any improvement in their developmental success when compared to germ-free non-supplemented counterparts. Surprisingly though, we observed an impact on the sex-ratio of the resulting adults, with a significant enrichment of female mosquitoes emerging from B vitamin-treated larvae. We further investigate potential mechanisms involved in this sex-specific effect of B vitamins via transcriptomics, diet supplementation and bacterial monocolonisation. We show that during larval development, males require less bacteria-derived products, notably biotin, than females and that they are sensitive to lower doses of biotin, resulting in the observed sex ratio distortion.

## Results

### Vitamin supplementation affects mosquito sex-ratio in axenic conditions

We previously reported that *Ae. aegypti* larvae cleared of their microbiota at the third instar showed lower developmental success compared to conventionally reared larvae, with only 10–20% of individuals successfully completing the metamorphosis stage. The proportion of germ-free individuals reaching the adult stage increased to ~50–60% if folic acid (vitamin B9) was supplemented to the larval diet (0.25–1.25 mg/mL), suggesting an important role of the microbiota in providing this B vitamin to larvae[4]. As the rescue was not complete, we wondered if other B vitamins potentially produced by the microbiota were involved in mosquito larval development and could further increase the proportion of larvae developing into adults. We produced germ-free third-instar larvae, following our transient colonisation protocol[4]: after egg sterilisation, we mono-colonized first instar larvae with the *E. coli* HA416 strain, auxotrophic for D-Ala and *m*-DAP, in the presence of these amino acids, and starved larvae from D-Ala and *m*-DAP shortly after reaching the third instar to turn them germ-free. We supplemented these larvae with a solution containing six B vitamins (biotin, folic acid, nicotinic acid, pyridoxine, riboflavin, and thiamine) and choline (not classified as B vitamin anymore, but recognised as essential nutrient), at concentrations previously reported to support *Ae. aegypti* larval development in germ-free conditions (VIT1×[13]). We did not observe any significant increase in the percentage of adult mosquitoes (Supplementary Fig. 1, proportion of adults: $p = 0.55$, not significant (ns); see Supplementary Data 2 for details on statistics). Since the folic acid concentration supplemented in our previous study was 5–25 times more concentrated that the concentration used in ref. 13, we decided to test the effect of four and eight times higher B vitamin concentrations (VIT4× and VIT8×) on larval development. Again, we did not observe an increase in the proportion of adult mosquitoes, rather a marginally significant decrease with VIT8× (Fig. 1A, proportion of adults: $p < 0.0001$; AUX vs GF, VIT4×, or VIT8×: $p < 0.0001$; GF vs VIT4×: $p = 0.73$; GF vs VIT8×: $p = 0.058$; Supplementary Data 2). However, under germ-free conditions, the small number of mosquitoes that completed their development was due to an equal proportion (~30%) of larvae either dying or being stalled at the larval stage. In contrast, the vitamin treatment significantly increased mortality rates, doubling them to approximately 60% (Fig. 1A, proportion of dead larvae: $p < 0.0001$; AUX vs GF, VIT4×, or VIT8×: $p < 0.0001$; GF vs VIT4×: $p = 0.010$; GF vs VIT8×: $p < 0.0001$; Supplementary Data 2). Interestingly, we saw a significant shift in the sex ratio of the fully developed mosquitoes, with more than 80% of adult mosquitoes being females when larvae were treated with VIT4× and VIT8× solutions (Fig. 1B, $p < 0.0001$; AUX or GF vs VIT4×: $p = 0.0010$; AUX vs VIT8×: $p = 0.0037$; GF vs VIT8×: $p = 0.0019$; Supplementary Data 2). This suggested a sex-specific effect of B vitamins on mosquito development.

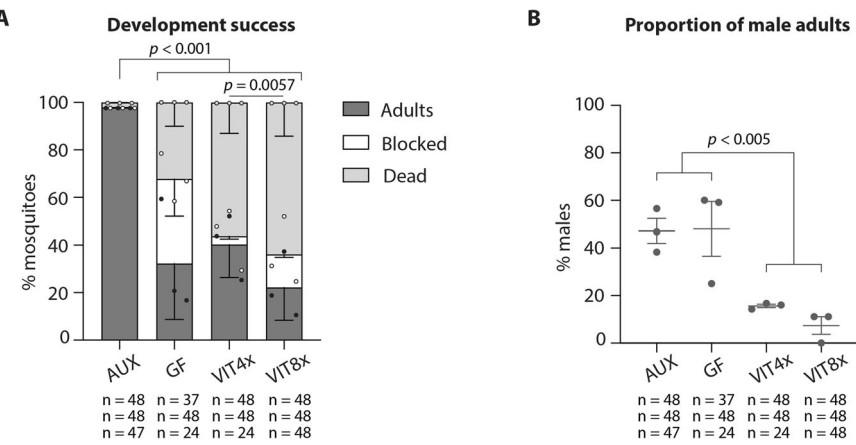

**Fig. 1 | Development success and sex-ratio of mosquitoes exposed to B vitamins during larval development. A** Proportion of individuals completing their development (dark grey), blocked at the larval stage (white), or dying during development (light grey) when initially reared with the auxotrophic *E. coli* strain (AUX), reversibly colonized at the beginning of the third instar and kept thereafter in germ-free conditions (GF) or supplemented with a 4× (VIT4×) or 8× (VIT8×) B vitamin solution. Bar charts represent the mean ± SEM of three independent replicates (individual points). **B** Proportion of male adult mosquitoes emerging from the larvae treated in (**A**). Numbers below graphs indicate the number of mosquitoes analysed per replicate and condition. See Supplementary Data 2 for detailed statistical information.

## Transcriptomic analysis of germ-free larvae treated with B vitamins

We had two alternative hypotheses to interpret the observed impact of B vitamins on sex ratio without significantly affecting overall development success. This effect can be attributed to a concomitant increase and decrease in the fitness of female and male larvae respectively upon vitamin supplementation, due to differential nutritional needs and to toxic effects of these compounds at the tested concentrations. Alternatively, some feminisation mechanism may influence male trait maturation during larval development. To sort out which of the two possibilities prevailed, we conducted a transcriptomic study on male and female larvae cleared of their microbiota at the beginning of the third instar, kept in germ-free conditions (GF), and then supplemented with VIT8× until sampling, 3–6 h after the beginning of the fourth larval instar (Fig. 2A; this transcriptomic study is detailed in Supplementary Note 1 and Supplementary Figs. 2 and 3). We observed on one hand that VIT 8× induced the heat-shock-protein related gene *AAEL017976* in males, which would lend credence to the toxicity hypothesis (Fig. 2B). In parallel, it down-regulated *AAEL012340* and up-regulated *AAEL017067* in females, assigned to encode lipase 1 precursor and peritrophin-48-like, which also may point to a better development in females (Fig. 2D, E). Conversely, we also observed a strong induction of *AAEL013606* in males, predicted to encode a SRY (Sex-determining region of Y chromosome)-like protein, which is a sex-determination factor and may alternatively point to a potential feminising impact of B vitamins (Fig. 2C). Hence, our transcriptomic analysis did not provide a clear conclusion on the way vitamins affected the sex-ratio.

## Effect of individual B vitamins on larval development and adult sex ratio

We reasoned that if vitamins had a toxic effect on males, we could detect it more clearly at a higher concentration. For this reason, we treated decolonised *Ae. aegypti* larvae with vitamins at high dose, 50× compared to the reference concentration[13], using single vitamins to narrow down which B vitamin was causing this sex-specific effect. We added a 16× dose of folic acid to all tested conditions to obtain enough adult mosquitoes in the germ-free control group and have a more reliable comparisons on adult mosquito sex-ratio. We observed a significant decrease in larval development success

in folic acid and biotin treatments (Fig. 3A, proportion of adults: $p < 0.0001$; GF vs folic acid: $p = 0.015$; GF vs biotin: $p < 0.0001$; all other comparisons $p > 0.05$, ns; Supplementary Data 2). In particular, while ~90% larvae developed into adults in the germ-free control, only ~60% and ~5% mosquitoes reached the adult stage in the folic acid and biotin treatments, respectively. While the addition of folic acid induced a non-significant increase in the proportion of mosquitoes that were blocked in their larval development (from ~2% in the control to ~ 20%, Fig. 3A, $p = 0.0002$; GF vs folic acid: $p = 0.062$, ns; Supplementary Data 2), biotin induced a strong mortality on larvae (from ~10% in the control to ~70%, Fig. 3A, $p < 0.0001$; GF vs biotin: $p < 0.0001$; Supplementary Data 2). The sex-ratio of the resulting adults was not significantly affected by any vitamin supplementation, although no viable male mosquito developed from biotin-treated larvae (Fig. 3B, $p = 0.73$, ns; Supplementary Data 2). This low statistical effect was probably due to the small number of biotin-treated larvae that could complete their development (adult mosquitoes per replicate (sex): 1/24 (non identified); 1/24 (female); 3/24 (females)), suggesting that the biotin 50× dose was generally toxic to both sexes. However, we could not determine if the absence of fully developed males after biotin supplementation was due to a male-killing effect or to the feminisation of phenotypic characters in male mosquitoes.

## Biotin requirements and toxicity are sex-specific

The experimental set up used in previous experiments could not distinguish between biotin-related male toxicity or feminisation of male mosquitoes. In fact, our visual analysis of fully developed mosquitoes indicated that adult mosquitoes displayed female-specific phenotypic characteristics, but this analysis did not yield information on the genotype of those mosquitoes and of those that did not reach adulthood. We took advantage of a recently established *Ae. aegypti* genetic sexing strain (Aaeg-M) characterised by the insertion of the *eGFP* transgene in the male-specific M locus[14]. This allowed us to sort first instar larvae by sex right after egg sterilisation and perform the full experiment on male and female larvae in parallel. The effect of four increasing biotin concentrations (1×, 4×, 8× and 20×) was tested on decolonized third instar larvae. As done previously, we added 16× folic acid to all tested solutions to increase the number of adult mosquitoes.

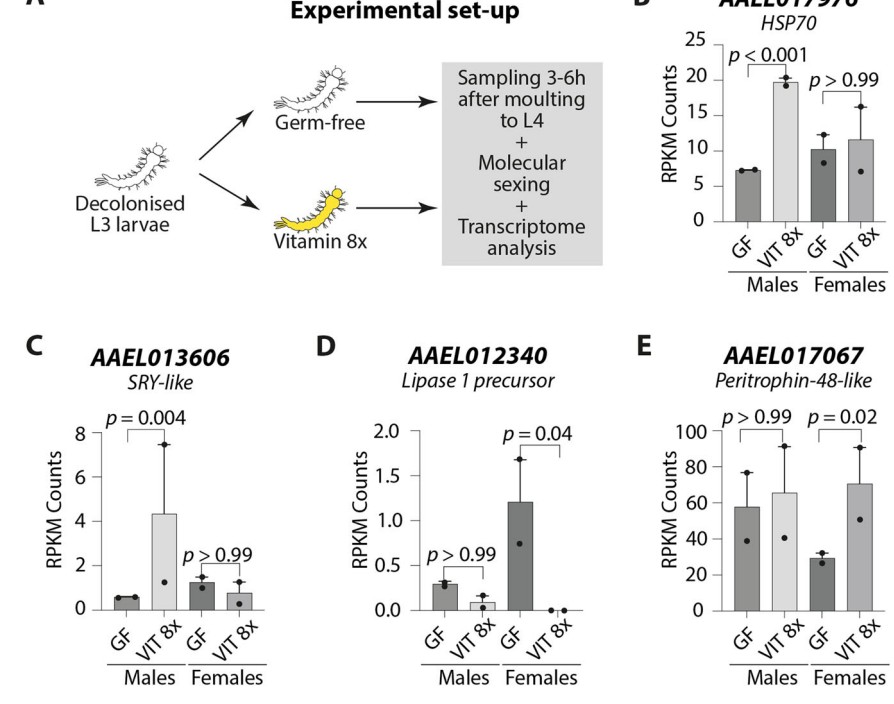

**Fig. 2 | Transcriptomic analysis of male and female larvae exposed to B vitamins. A** Experimental set-up: mosquito larvae reared in gnotobiotic conditions with the auxotrophic *E. coli* strain were decolonized at the beginning of the third instar (L3) to obtain germ-free larvae. Larvae were either kept in germ-free conditions or treated with 8× B vitamin solutions. After three to six hours since moulting to the fourth instar (L4), larvae were individually collected. DNA was extracted from individual larvae for sex assignment using the *Nix* gene, while RNA was extracted for transcriptome sequencing on male and female larvae. A more precise version of the experimental set up can be found in Supplementary Fig. 2. **B–E** Reads Per Kilobase per Million mapped read (RPKM) values of representative genes differentially regulated with B vitamins in males (**B, C**) and females (**D, E**). Bar charts represent the mean ± SEM. Genes are named according to their Vectorbase ID.

**Fig. 3 | Development success of male and female mosquito larvae exposed to high doses of single B vitamins or to increasing doses of biotin.**
**A** Proportion of mosquitoes completing their development (dark grey), blocked at the larval stage (white), or dying (light grey) when reversibly colonized at the beginning of the third instar and kept in germ-free conditions (GF) or supplemented with a 50× solution of folic acid, choline, thiamine, nicotinic acid, biotin, riboflavin, or pyridoxine. A 16× folic acid solution was also added to all tested conditions. Bar charts represent the mean ± SEM of three independent replicates (individual points) except for riboflavin (two replicates). **B** Proportion of male adult mosquitoes emerging from the larvae treated in (**A**). **C–E** Proportion of mosquitoes (Aaeg-M strain) completing their development (dark grey), blocked at the larval stage (white), or dying (light grey) when reversibly colonized at the beginning of the third instar and kept in germ-free conditions (GF) or provided with a 1×, 4×, 8×, or 20× biotin solution. A 16× folic acid solution was also added to all tested conditions. Bar charts represent the mean ± SEM of four independent replicates (individual points) except for GF + folic acid (three replicates). Data in (**C**) are not sorted by sex, while data in (**D**) and (**E**) represent results for male and female mosquitoes, respectively. Numbers below graphs indicate the number of mosquitoes analysed per replicate and condition. See Supplementary Data 2 for detailed statistical information.

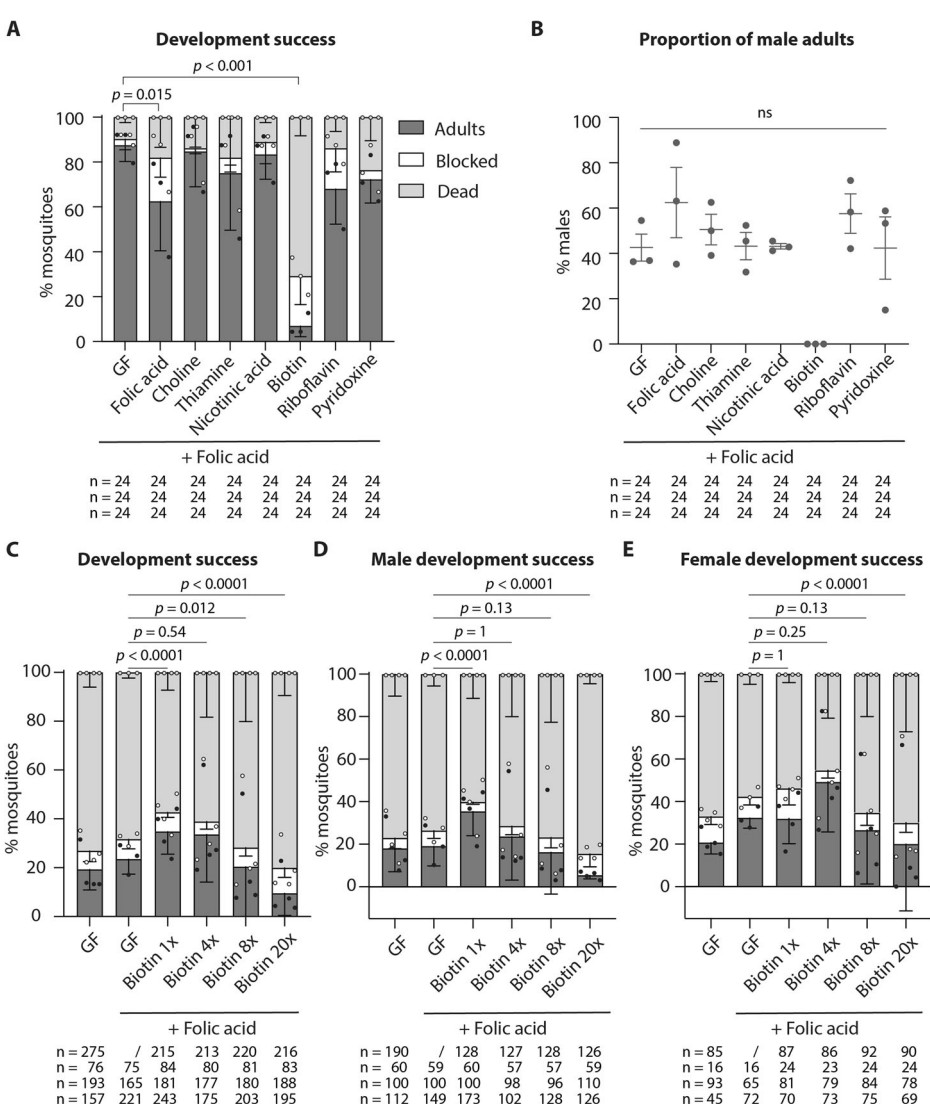

When analysing the global results independently from the mosquito sex, we observed that the addition of folic acid alone did not significantly increase the proportion of larvae completing their development to adulthood in the Aaeg-M strain, with ~20% of fully developed adults in both germ-free controls, with or without folic acid (Fig. 3C, $p < 0.0001$; GF vs GF + folic acid: $p = 1$, ns; Supplementary Data 2). The proportion of fully developed mosquitoes at 1× biotin concentration increased to ~30%, while it decreased to ~20% and ~10% in 8× and 20× biotin treatments, respectively (Fig. 3C, see Supplementary Data 2 for individual comparisons). In all treatments, the variation in the proportion of adult mosquitoes was due to an equivalent change in the percentage of dead mosquitoes rather than of stunted larvae.

Data sorted by sex indicated a differential biotin requirement for male and female mosquitoes: while the biotin 1× concentration was the best one to support the development of males (~30%, Fig. 3, $p < 0.0001$, see Supplementary Data 2 for individual comparisons), the 4× concentration resulted in the highest development success in females (~50%, Fig. 3E, $p < 0.0001$, see Supplementary Data 2 for individual comparisons). In females, the effect of the addition of biotin 4× was not statistically different from germ-free conditions supplemented with folic acid ($p = 0.25$, ns; Supplementary Data 2), but the addition of biotin 4× significantly improved the proportion of developed mosquitoes in the absence of folic acid ($p = 0.0001$; Supplementary Data 2). As discussed below, the sole addition of folic acid did not significantly improve the development of germ-free larvae

in these conditions ($p = 1$, ns; Supplementary Data 2). To confirm that the toxic effect was specific for biotin and not a combinatory effect between biotin and folic acid, we tested these vitamins separately. Indeed, a 50× treatment with biotin alone significantly reduced the proportion of developed mosquitoes, in both males and females (Supplementary Fig. 4A–C).

When comparing male and female mosquitoes in their developmental rates to pupa (i.e., the proportion of larvae starting metamorphosis by day), we observed similar proportions of mosquitoes starting metamorphosis in all conditions, with a higher percentage of males going through pupation in germ-free ($p < 0.0001$), biotin 1× ($p < 0.0001$) and 8× ($p = 0.02$) treatments (Supplementary Fig. 5A and Supplementary Data 2). However, a significant proportion of these pupae were not able to complete the developmental stage and died before emergence (Supplementary Fig. 5B). Interestingly, the addition of 16× folic acid alone to germ-free larvae differentially affected the pupation rates of male and female mosquitoes, suggesting that sex-dependent requirements are not exclusive for biotin ($p = 0.0005$; Supplementary Fig. 5B and Supplementary Data 2). Taken together these data suggest that the clearance of the microbiota during the third instar impacts mosquito development at the metamorphosis stage, and that the addition of biotin at a 1× concentration improves male development while a 4× concentration already causes toxicity; in females, the optimal concentration for development is 4×. Since toxicity might be generally correlated to body size, we tested whether male larvae had a significant lower mass or size compared to female larvae, similarly to what observed in adult mosquitoes. Third instar

**Fig. 4 | Development success and sex ratio of larvae with decreased support in bacterial metabolites.** In A–D, auxotrophic *E. coli*-colonized larvae were treated with 1× or 0.01× concentrations of D-Ala and *m*-DAP to slow bacterial development. **A** Number of colony forming units (CFU)/mL in wells containing individual larvae treated with 1× (black dots) or 0.01× (white dots) D-Ala and *m*-DAP. Each dot represents a well, 6 wells/time point/replicate were tested in at least 3 replicates. Bars represent the mean ± SEM. Time-points indicate the time after bacteria were added to germ-free larvae. **B** Proportion of mosquitoes reaching adulthood (dark grey), blocked in development (white) or dead (light grey) when auxotrophic *E. coli*-colonized larvae were supplemented with standard (1×) or diluted (0.01×) D-Ala and *m*-DAP concentrations. Bar charts represent the mean ± SEM of five independent replicates (individual points). **C** Proportion of males amongst the adults and **D** number of days until reaching adulthood when auxotrophic *E. coli*-colonized larvae were supplemented with standard (1×, black) or diluted (0.01×, white) D-Ala and *m*-DAP. Bars represent the mean ± SEM. In (E, F), larvae were colonised with wt *E. coli*, the growth deficient mutant Δ*mnmA* or the biotin-deficient Δ*iscUA* mutant. **E** Proportion of mosquitoes reaching adulthood (dark grey), blocked in development (white) or dead (light grey) when larvae were colonised with different *E. coli* mutants. Bar charts represent the mean ± SEM of five independent replicates (individual points). **F** Proportion of males amongst the adults when larvae were colonised with different *E. coli* mutants. Bars represent the mean ± SEM. Numbers below graphs indicate the number of wells/mosquitoes analysed per replicate and condition.

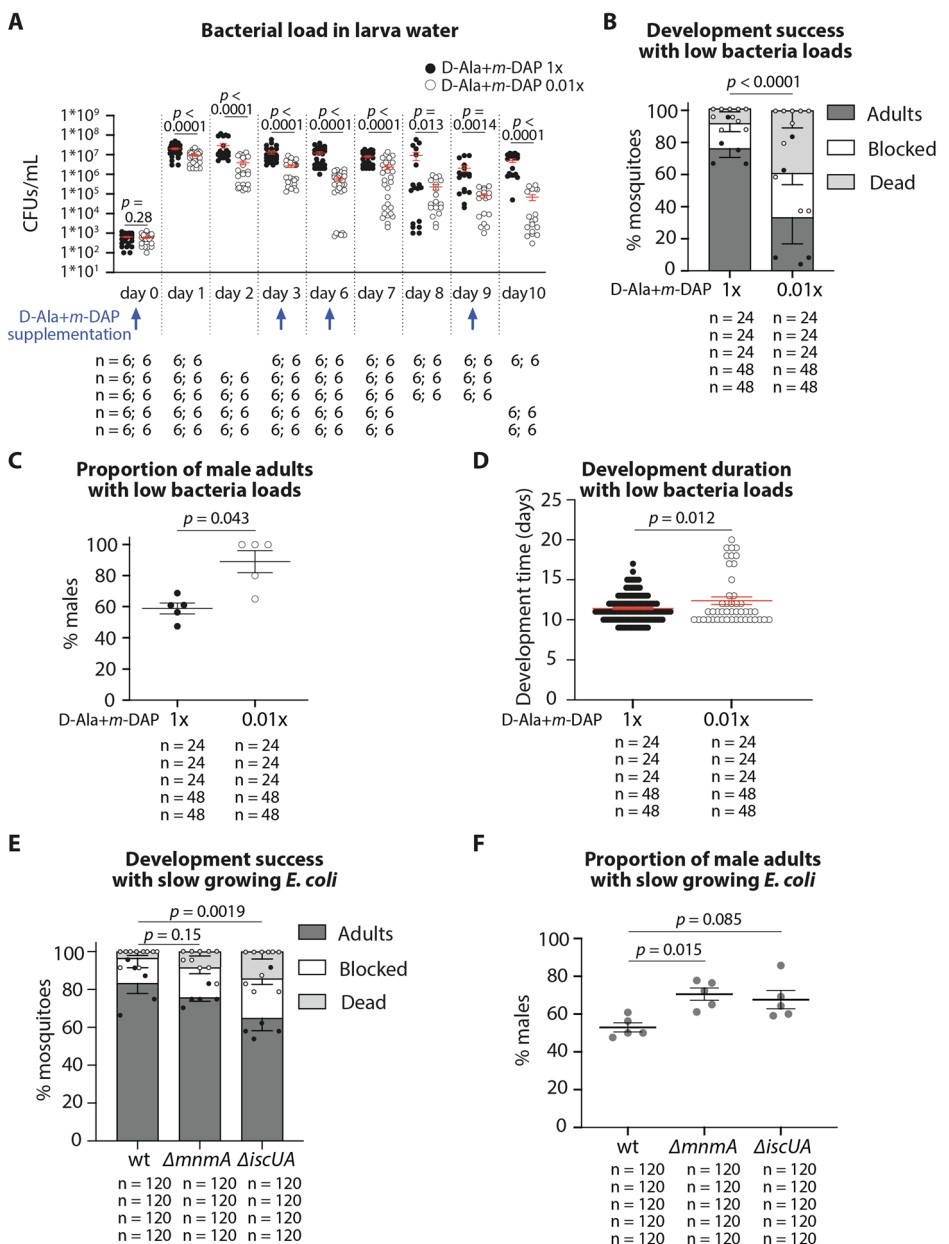

larvae did not show differences between sexes when measuring total dry mass or larval length (dry mass: $p = 0.13$; body length: $p = 0.54$, Supplementary Fig. 6A, B). However, male larvae had significant smaller head capsules (a proxy of body size) compared to females ($p = 0.01$, Supplementary Fig. 6A). Given that the weight difference between males and females is less than 6% and the difference in head capsule size is less than 2%, we conclude that the variations in biotin toxicity are likely attributable to other sex-specific factors beyond size.

For each experiment, adult mosquitoes were visually inspected to confirm that their genotype (GFP fluorescence status in L1) matched their adult phenotype. In parallel, an experiment with New Orleans mosquitoes was performed only on decolonised larvae kept germ-free or supplemented with 4× or 8× biotin. To compare mosquito phenotype and genotype, emerged mosquitoes were visually analysed to confirm their sex. Their DNA was extracted and subjected to dual PCR on *Nix* and *Actin*. PCR amplicons were analysed using a capillary electrophoresis system that allowed to automatically detect gene-specific signals. Among the analysed mosquitoes (GF: $n = 70$, biotin 4×: $n = 65$, biotin 8×: $n = 59$), none showed discordant

results between genotype and phenotype, further confirming that biotin had a toxic effect but did not induce any male feminisation.

## Males require less bacteria-derived metabolites than females for development

Based on these results, we hypothesised that males required lower quantities of bacteria-derived metabolites than females. To test this, we provided larvae with a suspension of auxotrophic bacteria approximately $10^6$ times more diluted than the standard solution used to support conventional mosquito development. This setup required active bacterial growth during larval development, leading to a greater dependency on the amino acids *m*-DAP and D-Ala for bacterial proliferation and larval development. Thus, a *m*-DAP and D-Ala solution at a conventional (1×) or 100-fold-diluted (0.01×) concentration were provided every three days until pupa appearance. This allowed to maintain a proliferating bacterial population while ensuring that bacterial loads were 2.2–73 fold lower when provided the diluted solution (Fig. 4A, 1× vs 0.01×: day 1, day 2, day 3, day 6, day 7, day 10: $p < 0.0001$; day 8: $p = 0.013$; day 9: $p = 0.0014$; Supplementary Data 2). Larvae received the

same amount of larval food in both conditions. This reduction in bacterial counts in larva water led to a lower development success (76% to 33%, $p < 0.0001$, Fig. 4B). Development was also slower (11.4 days to 12.4 days: $p = 0.012$; Fig. 4D and Supplementary Data 2). While females generally develop slower than males, this did not explain slower development as larvae that developed in 15 days or more in the 0.01× condition were only males. As expected, sex ratio was also distorted towards a higher proportion of males with lower amounts of bacteria (Fig. 4C; 59% to 89%, $p = 0.043$; Supplementary Data 2). When doing a tenfold dilution (0.1×), we observed similar trends, albeit with non-significant differences on sex ratio, partly because less replicates were performed (Supplementary Fig. 7A–D; % adults: $p = 0.0061$; sex: $p = 0.32$; time: $p = 0.012$; Supplementary Data 2).

We then wished to test whether such effects were specific to the provision of biotin. The use of a Biotin synthase (BioB) auxotrophic mutant was impossible in the experiment condition set-up because it would require the addition of biotin for its growth. We thus relied on a ΔiscUA mutant that is deficient in the main Fe-S cluster biogenesis pathway, hence predicted to be defective in the maturation of Fe-S proteins including BioB. BioB lacking this cluster was indeed reported to shows stability defect[15]. However, the ΔiscUA mutant is not auxotroph for biotin, likely because the second Fe-S biogenesis system SUF is taking over and providing some level of maturation of BioB. We observed that ΔiscUA has a significantly lower ability to support larval development to adulthood than the control (65% respect to 83%: $p = 0.0019$; Fig. 4E and Supplementary Data 2) and that it supported the development of a slightly higher proportion of males, albeit marginally significantly ($p = 0.085$; Fig. 4F and Supplementary Data 2). This result supported the view that biotin level lowering bears consequence on larval growth and sex ratio. Another possibility is that the reduced efficiency of the ΔiscUA mutant in supporting larvae development may be due to its slower growth rate and compromised fitness. Therefore, to test this last possibility, we used the ΔmnmA E. coli mutant known to show slow growth rate[16] (Supplementary Fig. 8). The gene mnmA encodes an enzyme required for thiolation of a subset of tRNAs. Larvae fed with ΔmnmA had an intermediate development success ($p = 0.15$; Fig. 4E and Supplementary Data 2) and a significant increase in proportion of males among adults ($p = 0.015$; Fig. 4F and Supplementary Data 2). Altogether these results indicated that microbial growth, and presumably associated richness in metabolite production, rather than specifically biotin level, is important for sex ratio.

## Discussion

The mosquito microbiota provides essential nutrients to its host. Among these nutrients, B vitamins have been shown to be required by mosquitoes to complete their larval development[4,6,13]. Here, we found that the larval B vitamins requirements, especially biotin, are sex-specific: males require less B vitamins than females for a successful development. Together, the differences in nutritional requirements and in thresholds of toxicity led to a sex-ratio distortion after supplementing diet of germ-free Ae. aegypti larvae with increasing doses of B vitamins. Specifically, there was a notable reduction in the proportion of male mosquitoes emerging from larvae that had been subjected to vitamin supplementation. We conclude that males require less bacterial-derived metabolites, including biotin, for development.

Evolutionarily speaking, favouring male development when nutritional conditions are scarce may be a way to maintain a progeny that will be successful if it spreads and finds mating partners elsewhere, while when nutritional conditions are optimal, having a high development success of both sexes may allow an efficient local colonisation. While our experiments were performed with individuals kept separately, the impact on sex ratio may be exacerbated in a population as male and female mosquitoes have different developmental dynamics at the larval stage, where females emerge later than males. In line with this, female adult body sizes are impacted more significantly than males by the composition of larval diet[17]. Female development may be longer because mosquito larvae must reach a critical mass to successfully commence metamorphosis, and this mass is higher for females than males, both at the larval and adult stages[18]. In species of scarab beetles

where males are larger than females, instantaneous growth rate is not different between sexes but optimal growth lasts longer in males[19]. Contrary to what we observe in mosquitoes, total development time is however not necessarily correlated with size in these beetles[19]. Similarly in Drosophila, females are larger than males, yet they develop on average 4 h earlier than males, indicating that final mass and development timing are not necessarily positively correlated. During fly metamorphosis, this protogyny phenotype is genetically controlled by Sex lethal, the master sex switch gene[20].

Diet composition at the larval stage has already been shown to affect the sex ratio of the adult mosquito population, with higher proportions of males emerging in starvation or suboptimal feeding conditions[21]. Our data now specifies that males and females have different requirements in bacteria-derived metabolites, notably biotin, while the amount of provided food did not vary in our experiments. Females may also need more energy storage than males for egg production, as egg production after the first blood meal has been found to be strongly correlated with richness of the larval food[22].

The observed male-killing effect appears to be a specific outcome resulting from exposure to one or a combination of B vitamins when exceeding a concentration threshold. We observed that the optimal concentration of biotin for the development of germ-free female larvae was four times higher than what had been previously reported as sufficient to sustain mosquito larval development in germ-free conditions[13], while this concentration was already toxic to males. Depending on the vitamin, our 4× and 8× concentrations were also 2–160-fold higher than the concentrations used more recently to formulate a chemically defined medium for germ-free Ae. aegypti larvae[6]. These two studies identified osmotic pressure as a critical limiting factor for artificial diets, describing a rapid larval mortality when the rearing medium contained either 11.6 g/L[13] or 113.7 g/L[6] of amino acids. The osmotic pressure of these solutions is ~25 or ~256 times higher than that of the 8× vitamin B solution we tested. Therefore, it is unlikely that the elevated mortality observed in our experiments could be due to high solute concentration. Yet, our transcriptomic analysis revealed the upregulation of a chloride channel encoding gene in both male and female larvae treated with the 8× vitamin solution, which may reflect a response to an increase in solute concentrations.

To investigate the potential mechanisms behind the male-specific toxic effect of B vitamins, we studied the transcriptomes of germ-free male and female larvae exposed to vitamin solutions. We encountered several problems in the process of sample sex assignment via PCR and RNA degradation, which led us to exclude one replicate and the entire vitamin 1× condition. This resulted in a relatively low number of genes that exhibited sex-specific regulation when compared to a prior transcriptomic study[23]. With the same sex assignment technique, Matthews et al.'s reported approximately 3400 differentially regulated genes between male and female fourth instar larvae. Between 35% and 40% of the genes identified in our transcriptomic study overlapped with those identified by Matthews et al., suggesting a reasonable degree of consistency with our data, considering that our larvae were germ-free. Most significantly, Nix was specifically enriched in male transcriptomes. These observations suggest that while we likely missed many regulated genes, those identified in our study are reliably true positives. The analysis of genes differentially regulated in vitamin 8× conditions identified different sets of genes in male and female larvae. Notably, the upregulation of genes linked to stress response, DNA binding, development or sexual reproduction was specific to male larvae. This observation suggests that the administration of high doses of vitamins interferes with critical cellular processes, as indicated by the upregulation of a heat-shock protein coding gene. The vitamin-induced stress response might explain the observed reduction in developmental rates among male larvae. However, it remains challenging to draw definitive conclusions regarding the precise mechanisms that underlie these effects.

When we administered individual B vitamins at elevated doses (50×), we observed that biotin, and to a lesser extent, folic acid, had lethal effects on germ-free larvae. However, only biotin was found to have an impact on sex ratio at this dose. Using the Aaeg-M GSS strain, we were able to distinguish the developmental success of male and female larvae independently and to

**Table 1 | List of the B vitamins tested with their reference and stock concentrations, and the solvent used for stock solutions**

| Vitamin | Reference concentration[13] | Solvent | Stock concentration |
|---|---|---|---|
| Biotin (B7) | 0.5 µg/mL | NaOH 0.1 M | 0.4 mg/mL |
| Choline | 25 µg/mL | Water | 50 mg/mL |
| Folic Acid (B9) | 30 µg/mL | NaOH 0.1 M | 50 mg/mL |
| Nicotinic Acid (B3) | 10 µg/mL | NaOH 0.1 M | 50 mg/mL |
| Pyridoxine (B6) | 10 µg/mL | Water | 50 mg/mL |
| Riboflavin (B2) | 20 µg/mL | NaOH 0.1 M | 10 mg/mL |
| Thiamine (B1) | 10 µg/mL | Water | 50 mg/mL |

validate the absence of a feminising effect caused by biotin. To maintain consistent experimental conditions, we added folic acid to all tested conditions. Surprisingly, this addition in the absence of biotin did not significantly affect the proportion of fully developed adults in germ-free conditions, in contrast to what we typically observe (ref. 4, Fig. 3A). This discrepancy could potentially be explained by differences in the mosquito strains used in the two sets of experiments (New Orleans and Aaeg-M GSS, which was created from mosquitoes sourced in Thailand[14]). However, we think that the variations in the timing between the two types of experiments were responsible for the different outcomes of folate supplementation. Due to the large number of larvae and to diet autofluorescence, first instar larvae of the Aaeg-M GSS strain had to be starved for 24 h while assigning the sex via GFP fluorescence before adding bacteria and food. Hence, larvae were initially in a deprived metabolic status which may have carry-over effects on later development. We hypothesise that folic acid alone is not sufficient to complement their requirements. Furthermore, experiments showed a noteworthy degree of variability of development success between replicates. While experimental conditions were extremely controlled, replicates were conducted with different batches of eggs produced in conventional conditions. This variability underscores the influence of various factors such as the maternal gonotrophic cycle or the age of the eggs on the outcomes of larval development. The quality of the blood on which mothers have fed and the quantity of eggs laid by each female could potentially influence the amounts of vitamins in embryos, consistent with observations that arthropod development is influenced by mother's age and diet[7,24,25].

Using the GFP-sexing strain, we showed that male and female mosquito larvae cleared of their microbiota exhibit different biotin requirements for their development (0.5 µg/mL for males and 2 µg/mL for females). These different biotin requirements and observed toxicity at higher doses might correlate with larval size, which tends to be larger in females than in males during the larval stage. Importantly, both sexes show heightened mortality rates during metamorphosis when exposed to higher biotin concentrations. Although more than 50% of the larvae successfully transitioned into pupae under all tested conditions, only a range of 10–40% emerged as fully developed adults (see Supplementary Fig. 5A). This observation suggests that exposure to elevated biotin levels during the third and fourth instars predominantly impacts pupae. Biotin has been found to be essential for intestinal stem cell mitosis in *Drosophila*[26], and is more generally involved in cell cycle progression[27], suggesting its importance during life stages with high cell proliferation rates. During metamorphosis, tissue remodelling increases cell proliferation, hence the number of biotin-responsive cells may be relatively high. Elevated biotin doses may specifically deregulate cell cycle at this stage and explain why we observe high pupal mortality. In line with our observation, previous studies underscored the essential role of biotin in *Ae. aegypti* larval development, but they also showed that high biotin concentrations induced mortality in pupae when larvae were reared in both axenic or conventional conditions[28,29]. Some degree of biotin toxicity has been observed in pupae of the flour beetle *Tribolium confusum* as well[30]. Furthermore, high biotin concentrations that still allowed *Ae. aegypti*

development were reported to reduce adult fertility; they notably caused follicle degeneration in females during the post-ovipositional period[29]. This effect of biotin on fertility seems to be conserved in other insect species such as the Mexican fruit fly *Anastrepha ludens*[31], the house fly *Musca domestica*[32] and the beetle *Dermestes maculatus*[33]. In these insects, females tended to lay fewer eggs when fed biotin-rich diets and eggs showed lower hatching rates. In *D. maculatus* the impact of biotin on embryogenesis appeared to be related to the binding of this vitamin to insoluble yolk proteins, resulting in reduced amino acid availability within the embryo[34].

Mosquitoes are incapable of synthesising B vitamins, thus relying on their diet and microbiota for the essential provision of these metabolites[4,6–8]. Notably, germ-free mosquito larvae can only develop if a rich diet is provided and if measures are taken to protect vitamins from light-induced degradation[5,6]. Several bacterial strains including *E. coli* are capable of rescuing mosquito development when introduced to germ-free larvae, while others such as *Microbacterium* are not[3,4]. This correlates with genomic data showing that *E. coli* possess complete metabolic pathways for B vitamin synthesis[7], while *Microbacterium* does not[3,35]. Our findings regarding biotin toxicity and previous data indicating biotin impact on insect fertility even raise the possibility that bacteria might interfere with mosquito sex ratio or, more generally, with mosquito physiology, by delivering high concentrations of some B vitamins. Intriguingly, genomic analyses of *Wolbachia* genomes have identified a riboflavin transporter gene to be associated with cytoplasmic incompatibility[36]. Although vitamin concentrations used in our experiments differ significantly from what mosquito would encounter in natural conditions, our findings underscore the critical need to explore bacterial-induced mechanisms that involve B vitamins because they influence the mosquito host larval development, toxicity, fertility and the alteration of the sex-ratio.

## Methods
### Mosquitoes and bacteria
*E. coli* HA 416 was grown in lysogeny broth (LB) supplemented with 50 µg/mL kanamycin, 50 µg/mL *m*-DAP and 200 µg/mL D-Ala. For all experiments, *E. coli* cultures were inoculated from single fresh colonies in liquid LB with appropriate supplementation and incubated at 30 °C, shaking at 150 rpm for 16 h. *E. coli* FBE051 (wt), FBE356 (Δ*iscUA*) and FBE584 (Δ*mnmA*), used for Fig. 4D, E, were cultured in the same conditions in LB with (Δ*iscUA* and Δ*mnmA*) or without (wt) 50 µg/mL kanamycin supplementation.

*Ae. aegypti* mosquitoes belonged either to the New Orleans strain or to the Aaeg-M genetic sexing strain (GSS[14],). Both colonies were maintained under standard insectary conditions at 28–30 °C on a 12:12 h light/dark cycle. Mosquitoes are routinely blood-fed either on beef blood provided by the local slaughterhouse (Abattoir Territorial de Rémire Montjoly) or on anaesthetised mice. The protocol of blood feeding on mice has been validated by the French Direction générale de la recherche et de l'innovation, ethical board # 089, under the agreement # 973021. Gnotobiotic mosquitoes were maintained in a climatic chamber at 80% relative humidity on a 12:12 h light/dark 28 °C/25 °C cycle.

### Vitamin solutions
B vitamins (Sigma-Aldrich) were dissolved in the appropriate solvent at the concentration indicated in Table 1. Stock solutions were adjusted to pH 7.4–8.0 and filtered through a 0.22 µM membrane filter and stored at −20 °C until use.

### Generation of gnotobiotic larvae
Germ-free larvae were obtained as previously described[4]. Briefly, eggs were placed on top of a filtration unit and surface sterilised by subsequent washes in 70% ethanol for 5 min, 1% bleach for 5 min and 70% ethanol for 5 min. After rinsing three times with sterile water, eggs were transferred to a sterile 25-cm² cell-culture flask filled with ~20 mL of sterile water. Approximately 20–30 eggs were inoculated in 3 mL of liquid LB and incubated for 48 h at 30 °C shaking at 150–200 rpm to confirm sterility.

The following day (day two), a 16 h culture of auxotrophic *E. coli* was centrifuged, and the bacterial pellet was resuspended in 5 times the initial culture volume of sterile water supplemented with *m*-DAP (12.5 μg/mL) and D-Ala (50 μg/mL). Larvae were individually placed in 24-well plates together with 2 mL of bacterial suspension and ~50 μL of autoclaved TetraMin Baby fish food suspension. Larvae were kept in a climatic chamber at 80% RH with 12:12 h light/dark 28 °C/25 °C cycle.

To achieve bacterial decolonisation, larvae that moulted to the third instar in a time window of 5 h during day 4 were washed in sterile water and individually transferred in a 24-well plate filled with 1.5 mL of sterile medium and 50 μL of sterile fish food in each well. Larvae obtained with this method were shown to be largely germ-free after 12 h since transfer[4]. Routine checks were performed to detect bacterial contamination. Specifically, the water used to rinse larvae was plated in LB plates to verify the absence of contaminating bacteria, and in LB plates supplemented with *m*-DAP, D-Ala, and kanamycin, to check for the carryover of auxotrophic *E. coli*. Additionally, for all experiments monitoring larval growth, replicates showing high development rate were tested for bacteria contamination in later time-points and excluded if bacterial growth was detected.

The Aaeg-M GSS required a sex-sorting step prior to being transiently colonised by auxotrophic *E. coli*. This strain carries an *eGFP* transgene in the male-specific M locus, thus only male mosquitoes express GFP. Axenic first instar larvae were individually placed in 96-well plates and checked for green fluorescence under an EVOS FL Auto System (Thermo Scientific). After sex-sorting, larvae were individually placed in 24-well plates. Bacteria were added the following day (day three) therefore the experiment schedule was shifted by one day.

For experiments shown in Fig. 4 and Supplementary Fig. 7, bacterial suspensions were diluted in sterile water to 100 CFU/mL (i.e., approximately $10^6$ times more diluted than in the normal setup). In Fig. 4A–D and Supplementary Fig. 7, amino acids were provided at a conventional concentration (1×, *m*-DAP – 12.5 μg/mL and D-Ala – 50 μg/mL) or after 1:10 or 1:100 dilution in sterile water. For each condition, the initial amino acid concentration was provided again every three days until pupae appeared.

### Vitamin treatment of axenic larvae
At the time of transfer, germ-free third instar larvae were randomly assigned to the different treatments and kept individually in 24-well plates. The larval medium consisted either of sterile water alone (germ-free condition) or sterile water supplemented with a single vitamin or with a mix of B vitamins (see Table 1 for reference B vitamin concentration). Sterile fish food was supplied to each larva. The 24-well plates were placed in a climatic chamber at 80% RH with 12:12 h light/dark 28 °C/25 °C cycle into plastic boxes covered with aluminium foil to prevent B vitamins from light-degradation[6].

### Analysis of axenic larvae developmental success in different vitamin conditions
Larvae treated with different B vitamin conditions were observed daily to track their developmental success. For each larva, the following parameters were recorded: day of moulting to the fourth instar, day of pupation, day of adult emergence, sex; in case of failed development to adult, the day of death was recorded. Larvae were monitored for 14 days after transfer in the germ-free medium. If at the end of this period larvae were still alive but did not undergo metamorphosis, they were marked as "blocked".

### Transcriptomic analysis
All methodological details about transcriptomics are described in Supplementary Information.

### Dry mass, larval length and head capsule width measurements
Gnotobiotic Aaeg-M GSS larvae were obtained as described above and monitored for their development. Larvae were sampled within 5 h after moulting to the third instar, sorted by sex checking green fluorescence, and either used to quantify body dry mass ($n = 50$ per sex/replicate) or to measure body length and head capsule width ($n = 20$ per sex/replicate).

Larvae collected to quantify dry mass were pooled, dried for a total of 2 h in a SpeedVac Concentrator plus (Eppendorf) using the following program: 90 min of vacuum, 2 h of pause, 30 min of vacuum. Body length and head capsule width were measured on larvae immobilized in 60% ethanol using a Leica M165 C stereo microscope (Leica Microsystems). Larval body length was measured as the distance between the anterior border of the head and the posterior border of the last abdominal segment, excluding the siphon. Head capsule was measured as the distance between the root of the two antennae. Three independent replicates were performed.

### CFU quantification
For each condition, 20 μL of breeding water from 6 different wells were sampled and each sample was transferred into 200 μL of sterile water. For each time point, sampling was randomly carried out among the wells whose larvae were at the most advanced stage. For CFU counts, bacterial suspensions were serially diluted in sterile water and 10 μL was spotted on LB agar plates, incubated at 30 °C and counted 24 h later.

### Growth curves
Bacteria were grown overnight in LB medium at 30 °C then diluted in LB at 1:50. Growth curves were produced by quantifying optical density at a wavelength of 600 nm every 30 min during 24 h in a FLUOStar Omega plate reader (BMG Labtech). Plates were shaken before each measurement. All the conditions were performed in triplicates (i.e., starting from three distinct colonies) and technical duplicates.

### Statistics and reproducibility
Graphs were created with GraphPad Prism (version 10.0.2). All experiments were replicated at least two times (the majority three times) with independent mosquito egg batches, bacterial cultures and vitamin solutions. Each replicate included at least 24 individuals/treatment. Statistical analyses were performed with generalised linear mixed models using the lme4, lmerTest and lsmeans packages in R (version 4.3.0). For categorical data (development success, sex ratio) the replicate was set as a random effect and an ANOVA was performed on a logistic regression (glmer). For quantitative data (CFU counts, duration of development, larval length measurements) an ANOVA was performed on a linear regression (lmer). To compare larval body weight a paired *t*-test was performed. For developmental rate analyses, a Log-rank (Mantel–Cox) test was performed in GraphPad Prism testing the incremental proportion of pupae or adults per day. Supplementary Data 2 details statistical information and number of individuals analysed per replicate. Numbers were rounded to two significant figures.

### Reporting summary
Further information on research design is available in the Nature Portfolio Reporting Summary linked to this article.

### Data availability
The raw RNA sequencing data underlying Fig. 2 and Supplementary Figs. 2 and 3 are available in the Sequencing Read Archive (SRA) under the Bioproject ID PRJNA1155297. Source data are provided in Supplementary Data 1.

### Code availability
The R code to run statistical test is available at https://doi.org/10.5281/zenodo.13742138[37].

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

## Acknowledgements

We would like to pay our gratitude to our colleague Jean-Géraud Issaly, who passed away in December 2022 and whose work was essential for mosquito rearing and colony maintenance in the lab. We thank Eric Marois (CNRS, INSERM, University of Strasbourg) for providing the Aaeg-M strain, and Siegfried Hapfelmeier (University of Bern) for providing the *E. coli* HA 416 strain. We also thank Stencey Fontenelle and Gabrielle Georgeon (Institut Pasteur de la Guyane) for technical assistance during experiments and Emmanuel Sechet for strain handling. This study is funded by the French Government's Investissement d'Avenir program, Laboratoire d'Excellence "Integrative Biology of Emerging Infectious Diseases" (grant ANR10-LABX-62-IBEID), and by ANR JCJC MosMi funding to M.G. (grant ANR-18-CE15-0007).

## Author contributions

Conceptualization, Validation – O.R., J.S.S. and M.G. Methodology: O.R., J.S.S., Y.E., F.B. and M.G. Software, Formal analysis– O.R. and J.S.S. Data Curation, O.R., J.S.S. and Y.E. Investigation – O.R., J.S.S., C.G., Y.E., P.F.I. and M.G. Resources, F. B. Writing – Original Draft, Visualization – O.R. and M.G. Writing – Review & Editing – O.R., J.S.S., C.G., F.B. and M.G. Supervision, Project administration, Funding acquisition – M.G.

## Competing interests

The authors declare no competing interests.

## Ethics approval

This study was conducted in French Guiana, an outermost region of France. All authors except FB were based in French Guiana when participating to the

research. None of the authors of this specific manuscript is originally from French Guiana, contrary to other publications of our laboratory. Our team is strongly involved in promoting science locally via outreach activities and training of local students.
