## [Transparent Peer Review file · Communications Biology]

Nutritional sex-specificity on bacterial metabolites during mosquito (*Aedes aegypti*) development leads to adult sex-ratio distortion

Corresponding Author: Dr Mathilde Gendrin

Version 0:

Reviewer comments:

Reviewer #1

(Remarks to the Author)

Romoli et al present an analysis of the effect of different B vitamins on the development of germ-free *Aedes aegypti* larvae. The authors use a novel system to clear *Ae. aegypti* larvae (*E. coli* m-Dap and d-Ala auxotrophs) and then assess how supplementation with different vitamin mixes and concentrations influences larval development. They find that higher concentrations of vitamins result in fewer males developing to adulthood. They then performed a transcriptome and identified a limited number of genes that show differential expression between the treatment x sex interaction. The authors then determine that biotin seems to be solely or primarily responsible for the observed decrease in survival of germ-free mosquitoes. When the authors treated larvae with concentrated biotin + folate, no males successfully developed. Using lower concentrations of biotin, the authors show a dose-response of survival/developmental success with increasing concentrations of biotin. Using their ability to control bacterial growth via supplementation with growth-limiting amino acids (d-Ala and m-DAP) the authors show that reduced bacterial growth is associated with reduced larval developmental success, but that this disproportionately affected female larvae, suggesting that females require more bacterially-derived nutrients than males.

The manuscript is well written and clear, and the experimental system is novel and allows for the testing of hypotheses that otherwise would not be able to be tested. While the results are interesting as an exercise in understanding how microbially derived nutrients influence host development, I feel that the authors need to make a strong case for the biological relevance of the study, as the concentrations of vitamins necessary to induce these phenotypes seem far outside of what the mosquitoes would ever experience in a non-lab environment, especially when B vitamins are often a limiting nutrient in animal diets.

I have a few questions for the authors about the manuscript as it is presented.

My first question is whether the authors took into consideration differences in larval body size that may assort in a sex specific way (male larvae being smaller than female larvae). Toxicity of a compound is often mass-specific - LD50 is usually expressed as amount of a compound per unit mass. If males are inherently smaller than females, it would be expected that male larvae are more susceptible to biotin toxicity due to their smaller size. While adult male *Ae. aegypti* are substantially smaller than females, I am not sure if male larvae are smaller than female larvae. However, this is an important variable that the authors could address by measuring the head capsule width as a proxy for total body size. The authors could then assess the effects of B vitamins on larval size, and determine if size is a stronger prediction of successful pupation and eclosion than sex.

My second question is about the methods regarding "germ free" mosquito larvae. Have the authors ever corroborated their culture-based determinations with a molecular/genetic (i.e., 16S PCR) approach? The authors use culture-based methods for assessing germ free status (p 19, line 6-7) and then later state that the larvae are "mostly" germ free (p 19, line 16), I am a little skeptical of how consistently the larvae tested were really germ free. Given the magnitude of the differences in some of the results, a few non-germ free mosquitoes could have made quite a difference. Doing this post-hoc and adding it to the supplemental materials would clarify this issue. Alternatively, since the authors used a ribodepletion method, looking in the transcriptome for evidence of bacterial RNA. Depending on the ribodepletion method and what probes were used, it may be possible to detect 16S sequences from *E. coli*; alternatively looking for highly expressed *E. coli* genes may indicate what

fraction of samples may have been sterile. The advantage of using RNA is that it is less likely to capture any residual DNA from dead bacteria in the gut.

For the lipase expression data – though the fold change might be large and significant, this is likely because there were extremely low reads mapped to the gene in one condition and a only a few mapped in the other. While this may be statistically significant, I question if a change in expression of this magnitude is biologically relevant.

If the authors are proposing that male-specific biotin toxicity is the primary mechanism behind the distorted sex ratio they observe, could they confirm this in normal, non-germ free mosquitoes?

Minor comments:

Page 6, line 5-7 – this sentence is awkward and confusingly worded.

Page 7, line 3 – “In parallel, it regulated...” was the expression higher or lower?

How much biotin is in the fish food? (p19, line 211)?

How was the fish food sterilized (p20 line 9)?

I appreciate that there are page limits, but the authors could have added a little more information about the transcriptome to the main methods section, even if just to reference that that information is in the supplemental data.

For the larvae that are “blocked” how long do they remain blocked? Indefinitely?

Kevin Vogel, PhD
Assistant Professor
The University of Georgia

Reviewer #2

(Remarks to the Author)

Using their auxotrophic-gnotobiotic model, they previously demonstrated that folic acid could partially rescue the development of their axenic mosquitoes. In this study they investigated the role of other B-vitamins in mosquito development. They found that increasing concentrations of B-vitamins were toxic to mosquitoes as others have already demonstrated. While none of the additional vitamins seemed to fully rescue normal development when combined with folic acid, the authors did find that biotin concentrations affected male:female sex ratios in emergent adults. To identify an answer for these observations the authors employed transcriptomics; however this data was inconclusive. Subsequently the authors performed additional studies with additional that revealed that the biotin concentration effect was not the result of feminization of the larvae and was clearly associated with sex-specific nutritional needs. Overall, this is a well-written manuscript that demonstrates how the microbiota can shape mosquito phenotypes indirectly through the production of B-vitamins. Specific Comments below.

1. Why didn't the authors try to rescue normal development with the B-vitamins in the absence of folic acid? I understand the thought is that folic acid only partially rescued and therefore if another B-vitamin were needed the effect would be additive. Maybe all B-vitamins can partially rescue similar to folic acid. If this were the case then one might start looking for other metabolic pathways or nutrients that are required to permit full rescue.
2. When applicable, the total number of mosquitoes analyzed in each treatment group in each experiment should be added to the figures above/ below the columns.
3. Fig4 The timepoints used for measuring cfus represent hours post what? Treatment? 24 hrs post emergence? Some other set point? This should be clarified in the figure legend.
4. The citation style is not consistent throughout. Most of the time numerical citations are used but in a few instances author names are used for citation purposes.
5. Typo In19 pg 2 Shouldn't be poisoning; should be posing.
6. Ln1 Pg15 Should read “Use of the Aaeg...”
7. Ln21 Pg 15 Should be consistent not consistently
8. Ln3 Pg17 should read “while others such as”
9. Ln17 Pg 19 Should read “...step prior to being transiently...”
10. Why is the nomenclature in figure 4 (dev, non-dev, dead) different from the other three figures (development, blocked, dying). Should stay consistent.
11. I appreciate how the authors included the transcriptomics data, primarily in the supplement, despite the data set being less than optimal. They discussed its limitations and didn't try to overstate any conclusions. This transparency is refreshing.

Reviewer #3

(Remarks to the Author)

The manuscript by Romoli et al. clearly demonstrates that there are sex-specific nutritional requirements of *Ae. aegypti* larvae during development and specifically that males require less bacteria-derived nutrients than females during development. Furthermore, variation in the amount of B vitamins alters adult sex ratios and the authors conclude this is through a toxic effect of B vitamins on males, and not through a feminizing mechanism. The experiments in this manuscript are well thought out, well controlled, and the interpretation of the data is sound.

Major comments: The data presented and explained in figure 4 is slightly confusing to follow and not adequately described in the abstract. It is hard to follow the different bacterial treatments and how to interpret them. Effort can be made to clarify and present this in a more concise manner.

Version 1:

Reviewer comments:

Reviewer #1

(Remarks to the Author)

The authors have sufficiently addressed my concerns.

Reviewer #2

(Remarks to the Author)

The authors did a thorough job addressing the reviewer's concerns and have further strengthened the manuscript. I have no additional comments.

Reviewer #3

(Remarks to the Author)

My comments were adequately addressed, no additional comments

Reviewer #1

Romoli et al present an analysis of the effect of different B vitamins on the development of germ-free *Aedes aegypti* larvae. The authors use a novel system to clear *Ae. aegypti* larvae (*E. coli* m-Dap and d-Ala auxotrophs) and then assess how supplementation with different vitamin mixes and concentrations influences larval development. They find that higher concentrations of vitamins result in fewer males developing to adulthood. They then performed a transcriptome and identified a limited number of genes that show differential expression between the treatment x sex interaction. The authors then determine that biotin seems to be solely or primarily responsible for the observed decrease in survival of germ-free mosquitoes. When the authors treated larvae with concentrated biotin + folate, no males successfully developed. Using lower concentrations of biotin, the authors show a dose-response of survival/developmental success with increasing concentrations of biotin. Using their ability to control bacterial growth via supplementation with growth-limiting amino acids (d-Ala and m-DAP) the authors show that reduced bacterial growth is associated with reduced larval developmental success, but that this disproportionately affected female larvae, suggesting that females require more bacterially-derived nutrients than males.

The manuscript is well written and clear, and the experimental system is novel and allows for the testing of hypotheses that otherwise would not be able to be tested. While the results are interesting as an exercise in understanding how microbially derived nutrients influence host development, I feel that the authors need to make a strong case for the biological relevance of the study, as the concentrations of vitamins necessary to induce these phenotypes seem far outside of what the mosquitoes would ever experience in a non-lab environment, especially when B vitamins are often a limiting nutrient in animal diets.

We thank reviewer #1 for his comments. Indeed, we agree that the setting used in our experiments is far from natural conditions. However, we still consider interesting these observations on sex-specific toxicity and requirements for mosquito development, which can be uniquely studied using our setup. We added a sentence in the discussion part (page 18, lines 2-3), to underline the limitations of our study.

I have a few questions for the authors about the manuscript as it is presented.

Following, answers to specific questions:

Q1. My first question is whether the authors took into consideration differences in larval body size that may assort in a sex specific way (male larvae being smaller than female larvae). Toxicity of a compound is often mass-specific - LD50 is usually expressed as amount of a compound per unit mass. If males are inherently smaller than females, it would be expected that male larvae are more susceptible to biotin toxicity due to their smaller size. While adult male *Ae. aegypti* are substantially smaller than females, I am not sure if male larvae are smaller than female larvae. However, this is an important variable that the authors could address by measuring the head capsule width as a proxy for total body size. The authors could then assess the effects of B vitamins on larval size, and determine if size is a stronger prediction of successful pupation and eclosion than sex.

A1. We thank the reviewer for this useful comment. Although we recognize that what the reviewer suggests is the most logical way to assess whether larval size is the better predictor for successful development, it was not possible to maintain sterility and at the same time measure the size of larvae that were subjected to the B vitamin treatment using our experimental setting. Therefore, we quantified dry mass, larval length and the width of larva head capsule of male and female larvae in gnotobiotic conditions at the third larval stage. While the dry weight and larval length did not differ statistically between the two sexes (Supplemental Figure S6A-B), the width of the head capsule was slightly bigger in female larvae (Supplemental Figure S6C). However, these differences are small in magnitude (less than 6 % of difference in dry weight and less than 2 % difference in head capsule size) compared to the observed difference in biotin toxicity. Therefore, we believe that other sex-specific factors are responsible for the differential vitamin requirements/toxicity. We added these data and this hypothesis to our manuscript (page 10, lines 9-17; page 16, lines 15-17).

Q2. My second question is about the methods regarding “germ free” mosquito larvae. Have the authors ever corroborated their culture-based determinations with a molecular/genetic (i.e., 16S PCR) approach? The authors use culture-based methods for assessing germ free status (p 19, line 6-7) and then later state that the larvae are “mostly” germ free (p 19, line 16), I am a little skeptical of how consistently the larvae tested were really germ free. Given the magnitude of the differences in some of the results, a few non-germ free mosquitoes could have made quite a difference. Doing this post-hoc and adding it to the supplemental materials would clarify this issue. Alternatively, since the authors used a ribodepletion method, looking in the transcriptome for evidence of bacterial RNA. Depending on the ribodepletion method and what probes were used, it may be possible to detect 16S sequences from *E. coli*; alternatively looking for highly expressed *E. coli* genes may indicate what fraction of samples may have been sterile. The advantage of using RNA is that it is less likely to capture any residual DNA from dead bacteria in the gut.

A2. In all experiments, the water from decolonized (= germ-free) larvae was routinely tested to check for bacterial growth by plating onto LB plates (general bacterial contamination) and LB plates supplemented with mDAP and D-Ala (carryover of auxotrophic bacteria). However, we decided to be conservative in the manuscript and state that larvae are “largely” germ-free, as it is not possible to test all individual larvae in all conditions.

Moreover, in our experience, as soon as a contamination was occurring, larval development was rescued and mosquitoes were developing at an almost standard rate, which is also an additional way to exclude non-germ-free samples from further analyses. We clarified these aspects in the method section (page 19, lines 16-21).

We are therefore confident that differences observed in replicates or independent experiments derives from general differences due to the use of distinct egg batches or from the mosquito genetic background.

Q3. For the lipase expression data – though the fold change might be large and significant, this is likely because there were extremely low reads mapped to the gene in one condition and a only a few mapped in the other. While this may be statistically significant, I question if a change in expression of this magnitude is biologically relevant.

A3. We agree with the reviewer that RPKM values are relatively low for the lipase gene. We decided to calculate a Z score to evaluate in which samples RPKM values are significantly different from the average. We found significant or almost significant *p* values only for female vitx8 samples (rep A: *p* = 0.0002; rep B: *p* = 0.13), while for all other samples *p* values are > 0.33. This suggest that female vitx8 RPKM values are significantly higher than the average RPKM values in all samples for the lipase gene. Therefore, we decided to keep this result in our manuscript.

Q4. If the authors are proposing that male-specific biotin toxicity is the primary mechanism behind the distorted sex ratio they observe, could they confirm this in normal, non-germ free mosquitoes?

A4. Unfortunately, as soon as bacteria are present in larval water, mosquitoes are developing at standard rate. We believe that the presence of bacteria masks any beneficial or toxic effect of vitamins because bacteria are metabolizing them. It is therefore impossible to test the effect of vitamins in non-germ-free conditions.

Minor comments:

Page 6, line 5-7 – this sentence is awkward and confusingly worded.

We reworded this sentence as following: “However, under germ-free conditions, the small number of mosquitoes that completed their development was due to an equal proportion (~30%) of larvae either dying or being stalled at the larval stage. In contrast, the vitamin treatment significantly increased mortality rates, doubling them to approximately 60 %.”

Page 7, line 3 – “In parallel, it regulated...” was the expression higher or lower?

For lipase 1 precursor expression is down-regulated by vitamins, while for peritrophin 48-like expression is up-regulated. We clarified this in the text.

How much biotin is in the fish food? (p19, line 211)?

We don't know the amount of biotin contained in the fish food. The only information provided by the supplier is the amount of vitamin A and D3.

How was the fish food sterilized (p20 line 9)?

As stated at page 20, line 11, the fish food suspension was autoclaved.

I appreciate that there are page limits, but the authors could have added a little more information about the transcriptome to the main methods section, even if just to reference that that information is in the supplemental data.

We added a paragraph in the method section reporting to supplemental material (page 22, lines 4–5).

For the larvae that are “blocked” how long do they remain blocked? Indefinitely?

We monitored larval development for 14 days after decolonization. If at that point larvae were still alive but did not undergo metamorphosis, we marked them as “blocked”. We added this information in the method section (page 22, lines 1–2).

Reviewer #2

Using their auxotrophic-gnotobiotic model, they previously demonstrated that folic acid could partially rescue the development of their axenic mosquitoes. In this study they investigated the role of other B-vitamins in mosquito development. They found that increasing concentrations of B-vitamins were toxic to mosquitoes as others have already demonstrated. While none of the additional vitamins seemed to fully rescue normal development when combined with folic acid, the authors did find that biotin concentrations affected male:female sex ratios in emergent adults. To identify an answer for these observations the authors employed transcriptomics; however this data was inconclusive. Subsequently the authors performed additional studies with additional that revealed that the biotin concentration effect was not the result of feminization of the larvae and was clearly associated with sex-specific nutritional needs. Overall, this is a well-written manuscript that demonstrates how the microbiota can shape mosquito phenotypes indirectly through the production of B-vitamins. Specific Comments below.

We thank Reviewer #2 for their comments. Here's a point-by-point response to their requests:

1. Why didn't the authors try to rescue normal development with the B-vitamins in the absence of folic acid? I understand the thought is that folic acid only partially rescued and therefore if another B-vitamin were needed the effect would be additive. Maybe all B-vitamins can partially rescue similar to folic acid. If this were the case then one might start looking for other metabolic pathways or nutrients that are required to permit full rescue.

As the reviewer suggests, our rationale for using folic acid alongside other B vitamins was to achieve a complete rescue of larval development. Since B vitamins play distinct roles in cellular processes and act as cofactors for different enzymes, we considered it unlikely that their effects on larval development would be redundant, meaning that different vitamins might not serve the same biological function. Since a full rescue was not achieved, it is clear that other nutrients are required by larvae to complete metamorphosis. Our decolonization method might be used in the future to identify such nutrients.

To make sure that the effect of biotin was specific to this vitamin rather than a combinatory effect with folic acid, we tested whether the same toxicity of the 50x concentrations was observed without folic acid. Indeed, we confirmed that the effect of biotin was independent from folic acid (new Supplemental Figure S4).

2. When applicable, the total number of mosquitoes analyzed in each treatment group in each experiment should be added to the figures above/ below the columns.

This information has been added to all figures.

3. Fig4 The timepoints used for measuring cfus represent hours post what? Treatment? 24 hrs post emergence? Some other set point? This should be clarified in the figure legend.

The hours displayed in the graph indicate the time after addition of bacteria to germ-free larvae. This information has been added in the figure legend.

4. The citation style is not consistent throughout. Most of the time numerical citations are used but in a few instances author names are used for citation purposes.

This issue has been solved.

5. Typo Ln19 pg 2 Shouldn't be poisoning; should be posing.

6. Ln1 Pg15 Should read "Use of the Aaeg..."

7. Ln21 Pg 15 Should be consistent not consistently

8. Ln3 Pg17 should read "while others such as"

9. Ln17 Pg 19 Should read "...step prior to being transiently..."

10. Why is the nomenclature in figure 4 (dev, non-dev, dead) different from the other three figures (development, blocked, dying). Should stay consistent.

Thank you for these suggestions, they have all be integrated in the manuscript.

11. I appreciate how the authors included the transcriptomics data, primarily in the supplement, despite the data set being less than optimal. They discussed its limitations and didn't try to overstate any conclusions. This transparency is refreshing.

Thanks!

Reviewer #3

The manuscript by Romoli et al. clearly demonstrates that there are sex-specific nutritional requirements of *Ae. aegypti* larvae during development and specifically that males require less bacteria-derived nutrients than females during development. Furthermore, variation in the amount of B vitamins alters adult sex ratios and the authors conclude this is through a toxic effect of B vitamins on males, and not through a feminizing mechanism. The experiments in this manuscript are well thought out, well controlled, and the interpretation of the data is sound.

Major comments: The data presented and explained in figure 4 is slightly confusing to follow and not adequately described in the abstract. It is hard to follow the different bacterial treatments and how to interpret them. Effort can be made to clarify and present this in a more concise manner.

We thank Reviewer #3 for this suggestion. We changed:

- Figure 4, to make it easier to read;
- the manuscript, to explain better the experimental setup and rationale behind Figure 4 (see page 11-12);
- the abstract, to explain better these results (page 2, lines 11-13).

We hope that these modifications improve the comprehension of the experiments and their interpretation.